# FasterDiT: Towards Faster Diffusion Transformers Training without Architecture Modification

**Jingfeng Yao, Cheng Wang, Wenyu Liu, Xinggang Wang**[*]
School of EIC, Huazhong University of Science and Technology
Wuhan 430074, China
{jfyao, wangchust, wyliu, xgwang}@hust.edu.cn

## Abstract

Diffusion Transformers (DiT) have attracted significant attention in research. However, they suffer from a slow convergence rate. In this paper, we aim to accelerate DiT training without any architectural modification. We identify the following issues in the training process: firstly, certain training strategies do not consistently perform well across different data. Secondly, the effectiveness of supervision at specific timesteps is limited. In response, we propose the following contributions: (1) We introduce a new perspective for interpreting the failure of the strategies. Specifically, we slightly extend the definition of Signal-to-Noise Ratio (SNR) and suggest observing the Probability Density Function (PDF) of SNR to understand the essence of the data robustness of the strategy. (2) We conduct numerous experiments and report over one hundred experimental results to empirically summarize a unified accelerating strategy from the perspective of PDF. (3) We develop a new supervision method that further accelerates the training process of DiT. Based on them, we propose **FasterDiT**, an exceedingly simple and practicable design strategy. With few lines of code modifications, it achieves 2.30 FID on ImageNet at $256 \times 256$ resolution with 1000 iterations, which is comparable to DiT (2.27 FID) but $7\times$ faster in training.

## 1 Introduction

With the advent of Sora [36], its foundational model, Diffusion Transformers (DiT) [37], has ignited extensive research interest. DiT is characterized by its remarkable flexibility and scalability, demonstrating exceptional capabilities in both image[37, 32, 16, 9, 7, 8] and video generation [31, 33, 17]. However, similar to the challenges faced with Vision Transformers [15], DiT is associated with high training costs. Its convergence rate remains slow, necessitating over 4700 GPU hours training on ImageNet generation tasks at 256 resolution [2]. This significant computational demand highlights the need for enhancing training efficiency for large-scale training.

One effective method for improving training is to modulate the Signal-to-Noise Ratio (SNR) distribution across different timesteps during training. Denoising generative models [22, 30] create a conversion from noise to data and methodically transfer noise into data as the timestep, denoted $t$, progresses. In this conversion process, the SNR gradually increases from zero to infinity. Given a generative process $x_t = \alpha_t x_\star + \sigma_t \epsilon$. Assuming that the input data $x_\star$ are ideally normally distributed with a variance of one, the SNR is typically defined as the ratio of variances $\frac{\alpha_t^2}{\sigma_t^2}$ [24, 14, 34, 40, 20]. During one training step, for a pair of input data and noise, we randomly select one $t$ for training. Modulating noise scheduling [35, 25, 10], loss weight [34, 20], and timestep sampling strategy [16]

---

[*]Corresponding author: xgwang@hust.edu.cn
[2]Testing with H800 GPUs

38th Conference on Neural Information Processing Systems (NeurIPS 2024).

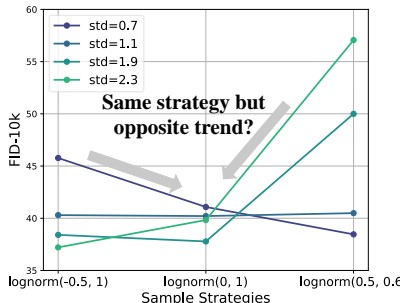

| Method | Arch. | Train Steps | FID-50k↓ |
|---|---|---|---|
| DiT [37] | | 400k | 19.5 |
| SiT [32] | DiT-XL-2 | 400k | 17.2 |
| **FasterDiT (Ours)** | | 400k | **11.9** (-5.3) |
| SiT [32] | DiT-XL-2 | 200k | 27.1 |
| **FasterDiT (Ours)** | | 200k | **17.5** (-9.6) |

Figure 1: (Left) **Problem Setting.** We find the same sampling strategy gets different performances with different data. (Right) **Performance of FasterDiT.** We improve Diffusion Transformers (DiT) training speed by a large margin without any architecture modification.

are common methods to modify the distribution of SNR during training. They are proven to be effective ways to improve training efficiency and effectiveness.

However, these methods do not always work across different data. For example, Stable Diffusion 3 (SD3) [16] proposes a timestep sampling strategy by logit normal function (lognorm) [1] for better training. Lognorm sampling is an excellent pioneering strategy and has been proven effective in subsequent work [17]. But it also has some limitations. According to SD3 exploration, $lognorm(-0.5, 1)$ substantially outperforms $lognorm(0.5, 0.6)$ in terms of Frechet Inception Distance (FID) results on ImageNet [13]. Nevertheless, we discover that this conclusion is a particular solution specific to the training data used. As shown in Figure 1, as we continuously reduce the signal strength, the FID results for $lognorm(-0.5, 1)$ progressively worsen, eventually becoming the poorest outcome. This underscores the necessity of evaluating the robustness of such methods across varying data conditions.

In this paper, we aim to provide a more comprehensive perspective for interpreting the problem. Our **first contribution** is to suggest interpreting the performance robustness with the perspective of the Probability Density Function (PDF) of SNR during training. Specifically, these methods typically modulate the distribution of SNR based on the timestep $t$. Although the same timestep has the same relative SNR as $\frac{\alpha_t^2}{\sigma_t^2}$, the absolute SNR at the step actually increases with the enhancement of the data signal [25, 10]. We believe that the original definition of SNR only reflects the relative signal-to-noise ratio in the same data distribution. However, the distribution of data-related absolute SNR during the training process is the key factor determining the effectiveness of the training. Hence, we slightly extend the definition of SNR and visualize its probability density function.

Our **second contribution** is to conduct extensive experiments and report about one hundred experiment results in our paper to empirically analyze the association between training performance and robustness with PDF. We analyze the differences in data robustness among commonly used basic pipelines on DiT [37, 32]. We find there is a trade-off between method robustness and performance, and suggest that the training process can be designed more intuitively from a PDF perspective.

The other method to improve training is to modify single-timestep prediction or supervision [34]. Our **third contribution** is the introduction of a new supervision method for velocity prediction-based approaches. With the same schedule, different prediction targets [34] or different supervised methods [22, 14] may also lead to different training results. Recently, velocity has been demonstrated to be a superior predictive target. Specifically, in addition to the traditional Mean Squared Error (MSE) loss, we have incorporated supervision of the velocity direction. We have found that this method is conceptually simple and significantly accelerates the training process.

To sum up, in this paper, we propose a new perspective on training the SNR PDF to offer a straightforward interpretation of the efficiency in training generative models (Section 2). We conduct extensive experiments and report near one hundred experimental results, aiming to empirically derive insights into training performance, robustness, and their correlations with the SNR PDF (Section 3). Subsequently, we apply these observations to refine the DiT process and, together with introducing a novel supervision method, developed FasterDiT to significantly enhance training efficiency (Section 4). FasterDiT achieves an FID of 2.30 on ImageNet at a resolution of 256, comparable to the original

DiT's FID of 2.27, yet achieves convergence seven times faster. We hope that our explorations contribute valuable insights to future research in generative model training.

## 2 Probability Density Function of SNR during Training

We hypothesize that the distribution of attention to different Signal-to-Noise Ratios (SNRs) during training is a crucial determinant of training efficiency and effectiveness. Here, two issues need to be addressed. Firstly, the previous definition of SNR as $\frac{\alpha_t^2}{\sigma_t^2}$ ignore the influence of data signal intensity, which is proven to be important during training [25, 35]. Secondly, a unified and intuitive approach is required to analyze the training SNR distribution. Therefore, in this section, we propose a slight modification to the existing definition of SNR. Subsequently, we utilize the Probability Density Function (PDF) of SNR during training to integrate noise scheduling, loss weighting, and timestep sampling strategies into a cohesive framework.

### 2.1 Preliminary

For ease of comprehension, we give a brief introduction to the formulation and training pipeline of generative models [22, 35, 29, 30]. Since our focus is on the SNR distribution, we universally consider flow matching and the diffusion model from a high-level perspective, similar to previous work [16]. Given data $x_\star \sim X$ and Gaussian noise $\epsilon \sim N(0, 1)$, we define the transport between noise and data as equation 1. In a discrete diffusion process [22, 35], $t$ is an integer from 0 to 1000. In a continuous flow process [29, 30], $t$ is a continuous value between $[0, 1]$. $\alpha_t$ and $\sigma_t$ is are coefficients related to $t$ defined differently according to schedules.

$$x_t = \alpha_t x_\star + \sigma_t \epsilon \tag{1}$$

During the training process, noised data $x_t$ and timestep $t$ are input to the generative model. The model is required to predict the specific target (noise, velocity, and so on). For example, in the most commonly used DDPM [22] pipeline, the prediction target is noise $\epsilon$ and the loss function is defined as equation 2. Only a single $x_t$ will be trained in each iteration for one image, instead of the whole timesteps. Therefore, some methods choose to change the sampling of $t$ [16] or give different loss weights [20, 35] according to $t$.

$$L_\theta = (\epsilon - \epsilon_\theta(x_t, t))^2 \tag{2}$$

### 2.2 Formulate Probability Density Function of SNR

**Data-dependent SNR** Previous work [24, 14, 34, 40, 20] highlights the importance of SNR. Typically, they assume that the data is an ideal normalization distribution and define the SNR as the ratio of variances (Equation 3). This definition is independent of data and only associated with the coefficients in timestep $t$.

$$\text{SNR}_{\text{prev}}(t) = \frac{\alpha_t^2}{\sigma_t^2} \tag{3}$$

However, the distribution of actual data is indescribable. It is closely linked to the nature of the image itself. In this study, we introduce a coefficient, $K(I)$, associated with the image $I$ that directly scales the signal. The value of $K(I)$ is influenced by various image characteristics such as the range of high and low frequencies, resolution, variance, and so on. It is known that the larger the variance of an image, the higher its SNR, illustrated as Equation 4. Further, for a specific dataset, we assume that the change in variance has little effect on the other nature of the image, i.e. $\text{std}^2$ is an independent variant in $K(I)$, and approximate $\frac{K(I)}{\text{std}^2}$ as a constant $C(I)$.

$$\text{SNR}(t) = K(I)\frac{\alpha_t^2}{\sigma_t^2}, K(I) \propto std^2 \tag{4}$$

$$\text{SNR}(t) = \frac{K(I)}{\text{std}^2}\frac{\text{std}^2\alpha_t^2}{\sigma_t^2} \approx C(I)\frac{\text{std}^2\alpha_t^2}{\sigma_t^2} \tag{5}$$

Our goal is to explore how variations in the SNR of different data lead to changes in training. However, constantly altering the training data is neither quantifiable nor convenient. The advantage of the above

**Algorithm 1** Estimate the Probability Density Function of SNR
___
1: Samples $N$ timestep $t$ uniformly
2: Calculate $f_l(t)$ and $f_s(t)$ for each $t$
3: Calculate $f_t(t) = \frac{f_l(t)f_s(t)}{\sum f_l(t) \sum f_s(t)}$ for each $t$
4: Estimate $f_t(t)$
5: Samples $N$ timestep $t$ from $f_t(t)$
6: Compute $\mathrm{SNR}(t) = C(I)\frac{\mathrm{std}^2\alpha_t^2}{\sigma_t^2}$ for each sampled $t$
7: Logarithmize $\mathrm{lgSNR(dB)} = 10\lg(\mathrm{C(I)}) + 20\lg\mathrm{std} + 10\lg\frac{\alpha_t^2}{\sigma_t^2}$
8: Estimate the probability density function of $\mathrm{SNR}(t)$
___

conversion (Equation 5) is that we can use the same dataset with different std to simulate different C(I) variations. This simplifies the scheme of probing the robustness of the data and allows us to not need to change the training data repeatedly.

**Weightings**  Consider a training process with the timestep sampling function $f_s(t)$ and loss weight function $f_l(t)$. For most cases, $f_s(t)$ and $f_l(t)$ are non-negative. To unify these into a single distribution of $t$, we define a new probability density function $f_t(t)$ as:

$$f_t(t) = \frac{f_l(t)f_s(t)}{\int_0^1 f_l(t)f_s(t)\,dt} \tag{6}$$

**PDF of SNR**  Now, we try to get the probability density function of SNR during training. It is independent with timestep $t$. Assume distribution $SNR(t) \sim Y$ and $t$ follows the distribution of $f_t(t)$. Solving for the distribution of $Y$ can be transformed into a problem of probability transformation. Mathematically, it could be defined as Equation 7.

$$f_Y(y) = f_t(g(y))\left|\frac{d}{dy}g(y)\right| \tag{7}$$

**Estimate of Probability Density Function of SNR**  In practice, the functions above are not always available. Hence, we use large amounts of discrete samples to approximate Equation 7. Here we provide its estimation algorithm. For all training processes, we visualize their PDFs in the way shown in Algorithm 1.

Now, we have developed a simple and feasible method for generating the PDF of SNR. It directly reflects the level of attention to different SNR values during the training process. Specifically, the higher the value of the PDF, the more frequently the corresponding SNR is trained during training. Next, we will conduct extensive experiments to analyze its relationship with different training data and training strategies.

## 3 What can we learn from SNR PDF?

Back to the problem we discuss in Section 1 Figrue 1. We argue that strategies that are solely based on $t$ tend to focus only on *relatively* high or low noise conditions in the generation process but are limited to considering the *absolute* Signal-to-Noise Ratio (SNR) with different signal intensity. They provide awesome solutions to improve the training but lack a comprehensive evaluation. In this section, we will explore this issue from the perspective of the Probability Density Function (PDF) of SNR, aiming to identify commonalities in these strategies and to delineate a more comprehensive design space.

### 3.1 Experiment Settings

We introduce our experiment settings first. We mainly adopt the Diffusion Transformer (DiT) training pipeline from two previous important works, DiT [37] and SiT [32]. We choose four different noise scheduling from them, including DDPM with linear beta schedule [22], DDPM with cosine beta schedule [35], flow matching with linear schedule (Rectified Flow) [30], and flow matching with cosine schedule [32].

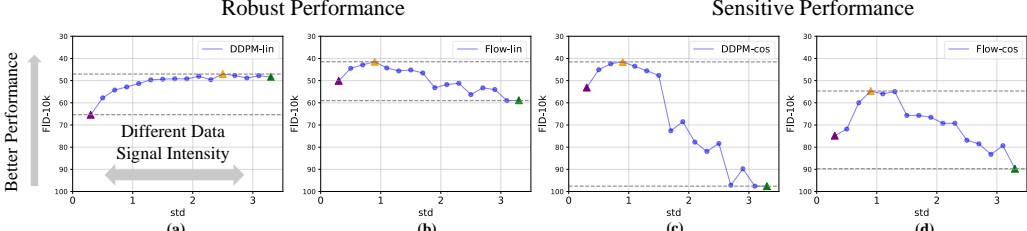

Figure 2: **Robustness of Different Noise Schedules.** By scaling input to different standard deviations, we compare the data robustness of four schedules [22, 35, 29, 32], including diffusion and flow matching. Note that we set the prediction target as noise for a fair comparison. We find that different data signal intensities lead to different generative performances and different schedules have different robustness.

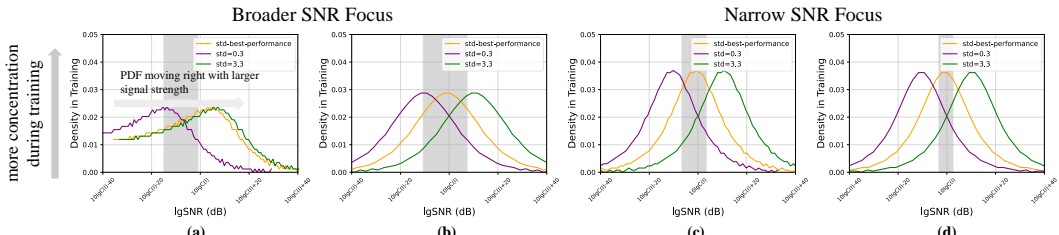

Figure 3: **SNR PDF of different noise schedules [22, 35, 29, 32].** The figure illustrates the signal-to-noise ratio (SNR) probability density functions (PDFs) for various schedules and standard deviations (see Section 2).

In the subsequent experiments, we initially set the prediction target of each noise schedule to noise (epsilon) to ensure a fair comparison. Subsequently, we modify the input strength by scaling the data to various standard deviations (std). We then shift the prediction target of flow matching to velocity the same as its original setting, to compare the similarities and differences between speed prediction and noise prediction outcomes. Each experiment was conducted on ImageNet [13] at a resolution of 128. We train each model for 100,000 iterations and assess their performance using the FID-10k metric for comparative analysis. Each experiment has been conducted with 8 H800 GPUs.

Our primary observational objectives are twofold: (1) Performance and robustness under different signal intensities. (2) Potential connections between the state of the PDF and performance and robustness.

### 3.2 Insights from PDF

**Different data signal intensities lead to different training effects.** In Figure 2, the prediction target for all schedules is uniformly set to noise to ensure a fair comparison. The curves demonstrate how different input intensities impact training outcomes. Notably, the performance of a single noise schedule fluctuates with changes in data intensity. For instance, the DDPM with a linear beta scheduler achieves an FID of 65.38 at $std = 0.3$ and improves significantly to 47.04 at $std = 2.5$ (Figure 2a).

**Different schedules exhibit significant differences in data robustness.** In Figure 2, as the intensity of the data varies, distinct schedules demonstrate significantly different robustness profiles. Specifically, with increasing standard deviation, the range of the FID for the DDPM-linear [22] schedule is observed at 18.03 (Figure 2a). In contrast, the DDPM-cosine [35] schedule exhibits a considerably larger FID range of 50.02(Figure 2c), highlighting substantial variability in performance between schedules.

**Schedule with a broad SNR focus could have a robust performance.** In Figure 3, we present the SNR PDFs as outlined in Section 2. For each noise schedule, we display three PDF curves corresponding to different signal intensities: low ($std = 0.3$), high ($std = 3.3$), and the level that results in the optimal FID. These PDF curves visually represent the training process's focus

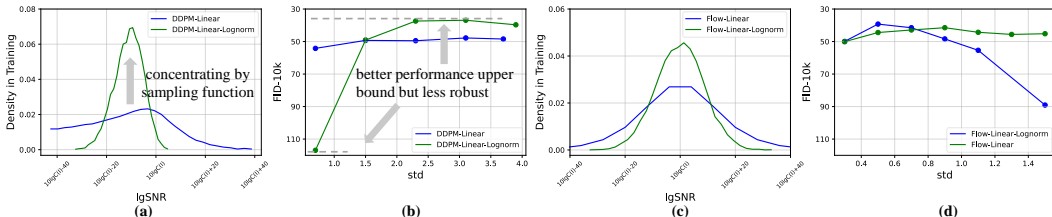

Figure 4: **Influence of Weghting Dring Training.** We use *lognorm*(0, 1) as Stable Diffusion3 [16]. The essence of this approach is to enhance the local focus of the PDF during the training process. This *increases the upper bound* of the training, but it also *reduces the robustness* of the training process to variations in the data.

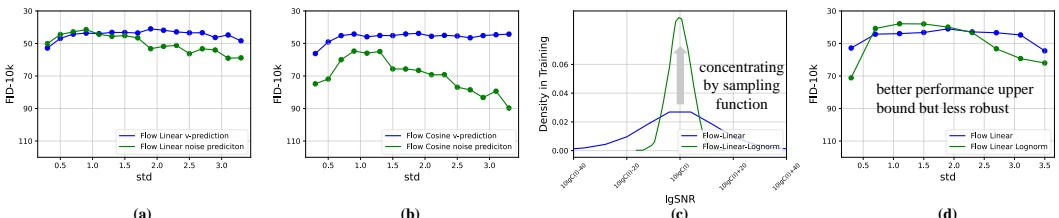

Figure 5: **Flow Matching with v-prediction.** We evaluate the robustness of commonly used flow matching [32] with v-prediction on ImageNet. (1) We find flow matching with v-prediction gets a more robust performance than noise-prediction. (2) There still exists a trade-off between performance and robustness.

across various noise interpeaksvals. Empirical analysis of the curve shapes across different noise schedules reveals that greater variance in the PDF, indicating a broader consideration of noise intervals (Figure 3a&b), is associated with improved stability in the schedule.

**The optimal SNR ranges for different schedules seem to be similar.** Direct observation of the standard deviations across different schedules does not readily lead to consistent conclusions. For instance, in Figure 2, while the DDPM-Linear schedule (Figure 2a) achieves optimal generative performance at an $std$ of approximately 2.5, the DDPM-Cosine schedule (Figure 2c) peaks at an $std$ of 0.9. However, insights might be gleaned from analyzing the PDF curves. In our experiments, as depicted in Figure 3, with a unified prediction target, better FID performance correlates with the mean of the PDF falling within the designated gray area. This observation offers a novel perspective and enriches our understanding of the previously established results.

**There is a trade-off between performance and robustness.** The primary purpose of employing weighting during the training process is to intensify the focus on specific SNR levels. We implement a logit normal function (lognorm) [1] to adjust the timestep sampling for two distinct schedules (Figure 2a&b), with results displayed in Figure 4. Our analysis reveals that while a well-defined sampling strategy can enhance performance, it may also introduce risks. For instance, the DDPM-Linear schedule achieves an optimal FID of 47.04, demonstrating robust performance across a range of 18.02. However, when we intensify the focus using lognorm (Figure 4a), although the best FID improves to 36.9, it also results in more variable outcomes, with a performance range widening to 80 (Figure 4b).

**Flow matching with v-prediction gets more robust performance.** Recently, flow matching [29, 30] has been regarded as a more concise and efficient pipeline for generative models. In the previous discussion, to facilitate a fair comparison, we set the prediction target on noise prediction. In Figure 5, We find flow matching with v-prediction achieves a more stable output performance, demonstrating enhanced robustness in data stability. For instance, Linear Flow exhibits a range in FID of 17.54 when using noise prediction, compared to 11.79 when using v-prediction (Figure 5a). This characteristic is even more pronounced in Flow with cosine schedule, where the ranges are 34.76 and 12.39, respectively (Figure 5b). Additionally, we find that the overall performance of v-prediction generally surpasses that of noise prediction.

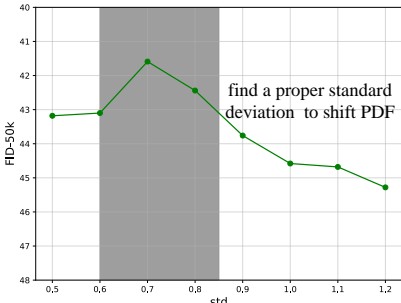

(a) **Standard Deviation Modulating.** Initially, we determine an appropriate range for the standard deviation.

| multi-step balance | velocity direction loss | FID-50k | | |
|:---:|:---:|:---:|:---:|:---:|
| | | 150k | 200k | 400k |
| ✓ | ✓ | **21.4** | **17.5** | **11.5** |
| | ✓ | 27.2 | 22.7 | 15.8 |
| | | 32.5 | 27.1 | 18.5 |

(b) **Ablations for FasterDiT Training.** Both multiple-step balance and velocity direction loss significantly enhance training efficiency.

**Algorithm 2** FasterDiT Training

```python
def lognorm(mu=0, sigma=1, size=None):

    # get logit normal distribution
    samples = scipy.norm.rvs(loc=mu, \
                scale=sigma, size=size)

    # transform to 0 to 1
    samples = 1 / (1 + np.exp(-samples))
    return samples

while training:
    # 1. data shifting
    # data: (B, C, H, W)
    # noise: (B, c, H, W)
    data = data * (target_std / data_std)
    noise = torch.rand_nlike(data)

    # 2. concentrating
    # timestep: (B, )
    t = lognorm(0, 1, t.shape[0])
    x_t = t*data + (1-t)*noise
    pred = model(x_t, t, y)

    # 3. improved supervision
    v = data - noise
    loss = (pred - v)**2.mean() + \
    1 - cosine_similarity(pred, v, dim=1).mean()

    loss.backward()
```

Figure 7: **Training Details.** Our training pipeline involves only minimal modifications to the code.

**The trade-off still exists in flow matching with v-prediction.** Furthermore, we observe that although v-prediction is generally more robust, there remains a trade-off between performance and stability. As shown in Figure 5c&d, when we used a lognorm to concentrate the focus of Linear-Flow, its performance resembled that of noise prediction, where the upper limit improved but stability decreased. We hypothesize that such trade-offs may be broadly prevalent.

To sum up, from the perspective of the PDF, efficient training requires satisfying two conditions: (1) the PDF should have a concentrated focus; (2) the focus area of SNR needs to fall within the correct range. Based on the simple observation, we try to improve DiT training.

## 4 Improving DiT Training

In this paper, our objective is not to devise novel model architectures for achieving state-of-the-art outcomes. Instead, we aim to explore a simpler, more interpretable, and universally applicable training approach for Diffusion Transformers (DiTs) [37].

In Section 3, we demonstrate a trade-off between performance and robustness in the DiT training process via signal-to-noise (SNR) probability density function (PDF) analysis. To expedite training, we should have a concentrated PDF that focuses on the right SNR during training. Initially, we select flow matching [30] with v-prediction as our noise schedule, owing to its robustness and superior performance. We then adjust the PDF to focus on the optimal SNR during training by modulating the standard deviation (std) of training data. Further, we employ a logit normal function [1] to sharpen the focus on the adjusted regions. Finally, we introduce a new, straightforward supervisory strategy that significantly enhances training outcomes.

### 4.1 Improving Multiple Step Balance

Here, we first modulate the standard deviation from 0.5 to 1.2 as illustrated in Figure 7 (left). During this sweep, we refrain from employing additional techniques such as weighting. It shows that when $std$ is around 0.70, the DiT training gets a better generative performance. Empirically, we set the target $std$ to 0.82, which is near the proper and stays consistent with previous work's setting [32, 37]. However, we argue that when the input data changes, e.g. different resolution, the choice of $std$ will

| Method | Model | Training Iters | FID↓ | sFID↓ | IS↑ | Prec.↑ | Rec.↑ |
|---|---|---|---|---|---|---|---|
| BigGAN [4] | BigGAN-deep | - | 6.95 | 7.36 | 171.4 | **0.87** | 0.28 |
| MaskGIT [6] | MaskGIT | 1387k×256 | 6.18 | - | 182.1 | - | - |
| ADM-*G* [14] | ADM | 1980k×256 | 4.59 | 5.25 | 186.70 | 0.82 | 0.52 |
| CDM [23] | CDM | - | 4.88 | - | 158.71 | - | - |
| RIN [26] | RIN | - | 3.42 | - | - | - | - |
| Simple Diffusion [25] | U-Net | 2000k×512 | 3.76 | - | - | - | - |
| Simple Diffusion | U-ViT-L | 500k×2048 | 2.77 | - | - | - | - |
| LDM-4-*G* [40] | LDM | 178k×1200 | 3.60 | - | 247.67 | 0.87 | 0.48 |
| U-ViT-*G* [2] | U-ViT | 300k×1024 | 3.40 | - | - | - | - |
| StyleGAN [42] | StyleGAN-XL | - | 2.30 | **4.02** | 265.12 | 0.78 | 0.53 |
| MDT-*G* [18] | MDT | 2500k×256 | 2.15 | 4.52 | 249.27 | 0.82 | 0.58 |
| DiT [37] | | 7000k×256 | 9.62 | 6.85 | 121.50 | 0.67 | 0.67 |
| SiT [32] | DiT-XL/2 | 7000k×256 | 8.61 | 6.32 | 131.65 | 0.68 | 0.67 |
| **FasterDiT** | | **1000k**×256 | 8.72 | 5.23 | 121.17 | 0.68 | 0.67 |
| | | **2000k**×256 | 7.91 | 5.46 | 131.27 | 0.67 | **0.69** |
| DiT *(cfg=1.5)* [37] | | 7000k×256 | 2.27 | 4.60 | **278.24** | 0.83 | 0.57 |
| SiT *(cfg=1.5)* [32] | DiT-XL/2 | 7000k×256 | 2.06 | 4.50 | 270.27 | 0.82 | 0.59 |
| **FasterDiT** *(cfg=1.5)* | | **1000k**×256 | 2.30 | 4.80 | 249.34 | 0.82 | 0.58 |
| | | **2000k**×256 | **2.03** | 4.63 | 263.95 | 0.81 | 0.60 |

Table 1: **Performance of FasterDiT on ImageNet 256×256.** Employing the identical architecture as DiT [37], FasterDiT achieves comparable performance with an FID of 2.30, yet requires only 1,000k iterations to converge.

differ. The strategy is similar to previous work [10], but we discuss it in a new perspective of training SNR PDF.

Then we improve the performance by a human-made concentration for PDF. Specifically, we use logit normal function [1] for timestep sampling, while *mu* and *sigma* are set to 0 and 1 respectively. Notably, *mu* is set to 0 because the PDF is already properly shifted (see Figure 3). It could help avoid instability discussed in Section 1 and improve the training performance (see Figure 7 (left) (b)).

## 4.2 Improving Single Step Supervision

Effective single-step supervision is essential in training generative models. For instance, employing different prediction targets can result in diverse predictive outcomes [34]. In the original DiT model, a strategy that simultaneously predicts noise and sigma is utilized to maximize performance [37]. Typically, the Mean Squared Error (MSE) loss function is employed for supervising these targets.

$$L_d = 1 - \frac{1}{HW} \sum_{h=0}^{H-1} \sum_{w=0}^{W-1} \frac{v_{gt}^{(h,w)} \cdot v_{pred}^{(h,w)}}{|v_{gt}^{(h,w)}||v_{pred}^{(h,w)}|} \tag{8}$$

Recently, in the context of flow matching [29, 30], the prediction target of velocity acquires a more specific physical significance, representing the flow rate from noise to data. Based on this, we hypothesize that supervising the direction of velocity could serve as an effective supervision strategy. Consequently, in this paper, we not only use the Mean Squared Error (MSE) to supervise velocity predictions but also apply cosine similarity to further supervise the directionality of velocity. Here we present velocity direction loss as shown in Equation 8, while $H$, $W$ denotes the height and width of predicted latent by DiT. We demonstrate that this combined supervisory approach significantly enhances the model's convergence rate (see Figure 7 (left) (b))

## 4.3 Comparison with Previous Methods

Owing to resource limitations, we train FasterDiT on ImageNet at 256 resolution [13] for 2,000k iterations and benchmark it against current advanced generative models (see Table 1). Our experiment

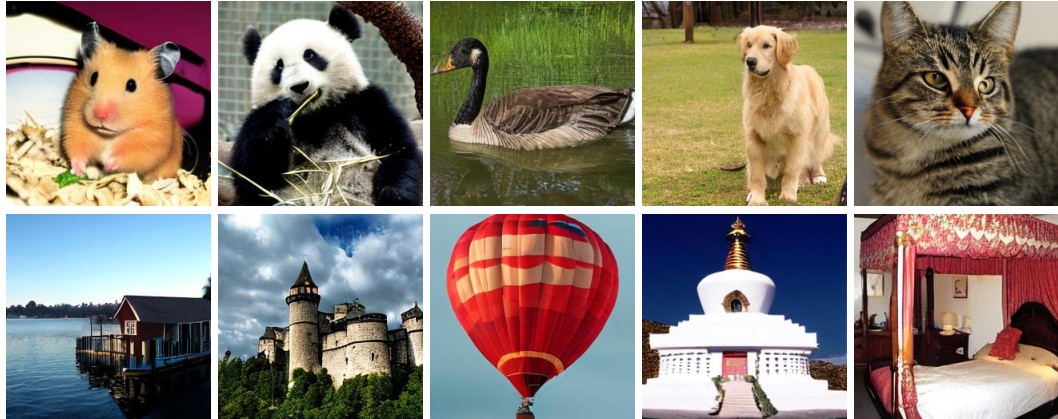

Figure 8: **Visualization Results.** We present visualization results for FasterDiT-XL/2 after training for 1,000k iterations, with CFG set to 4.0.

has been conducted with 8 H800 GPUs. Notably, the focus of our work is to explore training strategies beyond structural enhancements. Thus, FasterDiT utilizes the identical architecture as DiT [32]. As illustrated in Section 1 Figure 1, FasterDiT achieves an FID-50k score of 11.9 at 400k iterations, markedly outperforming the original DiT model (FID-50K 19.5), and its enhanced counterpart, SiT (FID-50k 17.2) [32]. Utilizing the same architecture as DiT [37], FasterDiT attains a similar performance level with an FID score of 2.30 but converges in just 1,000k iterations.

### 4.4 Performance on Higher Resolution Images

We evaluate FasterDiT on higher resolution generation tasks to explore its resolution generalization capabilities. Specifically, we apply our approach to DiT-B/2 and DiT-L/2 models for ImageNet generation at a $512 \times 512$ resolution. The results, presented in Table 2, indicate that FasterDiT consistently achieves faster convergence across all configurations. Notably, after 200k training iterations, our method improves the FID-10k performance of DiT-B/2 by 18.78 and DiT-L/2 by 17.93, underscoring its effectiveness for high-resolution image generation tasks.

### 4.5 Performance with Different Diffusion Architectures

We further apply our training method to other diffusion models beyond DiT, including Latent Diffusion Models [40] (using the UNet [41] architecture) and U-ViT [2]. Here, we specifically refer to the use of the previously mentioned multi-step balance and velocity direction loss. The results, presented in Table 3, indicate performance improvements for both U-ViT and UNet with our method. This suggests that our approach has the potential to generalize across a broader range of diffusion model architectures.

## 5 Related Work

### 5.1 Generative Models and Diffusion Transformers

The denoising diffusion probabilistic model [22] has gradually replaced GANs [12, 19] due to its stronger performance and more stable training, becoming the mainstream generative model. Among them, [14] first proves its effectiveness on ImageNet [13]. [40, 38] extend the diffusion process from the diffusion space to the space of VAE latent space, achieving high-performance, high-resolution generation. Meanwhile, some methods [32, 46, 17] have also trained generative models using flow-matching [29, 30] techniques to replace the diffusion path. These methods exhibit simpler mathematical properties and faster learning effects.

Among them, U-Net [41] is the most commonly used architecture for generative models. Recently, due to the scalability advantages of transformers [15, 44], some transformer-based [2, 37] generative

| Method | Models | Training Samples | Resolution | FID-10k |
|--------|--------|------------------|------------|---------|
| DiT | DiT-B/2 | 100k × 128
200k × 128 | 512×512 | 93.36
77.11 |
| FasterDiT | DiT-B/2 | 100k × 128
200k × 128 | 512×512 | 77.85
**58.33** |
| DiT | DiT-L/2 | 100k × 64
200k × 64 | 512×512 | 87.24
67.29 |
| FasterDiT | DiT-L/2 | 100k × 64
200k × 64 | 512×512 | 71.58
**49.36** |

Table 2: Performance on ImageNet 512×512.

| Model | Training Samples | FID-10k |
|-------|------------------|---------|
| U-ViT-L [2]
U-ViT-L + Ours | 200k × 128 | 50.22
**37.12** |
| LDM-UNet [40]
LDM-UNet + Ours | 200k × 128 | 66.73
**60.07** |

Table 3: Performance with Different Architectures.

models have also emerged. However, similar to Vision Transformers [15], Diffusion Transformers [37] converge slowly.

## 5.2 Fast Training of Generative Models

Work for accelerating Diffusion Transformers [37] (DiT) could be simply divided into two categories. One is architecture modification. MaskDiT [45] and MDT [18] combine mask image modeling [3, 21] pre-training and diffusion training for speeding up. Similarly, SD-DiT [46] takes one step further to combine DiT training with MoCo-like [11] contrastive learning. CAN [5] proposes a dynamic weights for condition to speed up training of diffusion models [37, 2].

Another strategy involves changing non-model design approaches. For example, using different noise schedules can achieve better results on images of various resolutions [25, 35, 10, 27]. Using different prediction targets, such as noise, data, or velocity, can also directly impact the effectiveness of the training [34]. Adjusting weights for the loss function and training sampling can also directly affect the outcomes of training [20, 16]. However, they constitute a vast design space with a high degree of direct interdependence, which makes the design of the model very complex.

## 6 Conclusion

**Limitations** We propose a novel approach called FasterDiT for accelerating the training of diffusion transformers. The main limitation of this paper lies in the lack of exploration of large-scale experiments, such as 2K high-resolution images, text-to-image generation, and video generation. Among these, we particularly focus on the text-to-image generation aspect. Specifically, in the class-conditional generation described in this paper, the DiT block only needs to process image tokens. However, in some text-to-image architectures, such as SD3 [16], self-attention needs to handle sequences combining text and visual tokens. The features from different sources (such as T5 [39] and VAE [28]) may lead to potential instability. We plan to further investigate this issue in the future.

**Societal Impacts** Our work has the potential to significantly enhance the efficiency of creative professionals in completing their creative processes. However, it also carries a risk of misuse.

**Conclusions** In this paper, we discuss training strategies to accelerate Diffusion Transformers. Initially, we slightly generalize the definition of SNR and conduct a unified analysis of various training strategies through the examination of SNR probability density functions during the training process. We find that there is a trade-off between robustness and performance in the training of diffusion models. Different noise schedules exhibit varying levels of robustness. Using weighting during the training can enhance the performance ceiling, but it may also decrease the robustness of the process. Additionally, we discover that utilizing a directional loss as an auxiliary loss function for velocity prediction significantly boosts training performance. Based on these observations, we conduct experiments with DiT-XL-2 on ImageNet 256×256 and observe substantial acceleration in training. We refer to this straightforward training approach as FasterDiT. We hope our work inspires further exploration into training strategies for generative models.

## Acknowledgments and Disclosure of Funding

This work was partially supported by the National Natural Science Foundation of China (NSFC) under Grant No. 62276108. We also extend special thanks to Bencheng Liao for his valuable suggestions.

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

# A Implementation Details

In this section, we offer a detailed overview of the experimental methodologies employed in our study. The investigation is delineated into two primary segments. Initially, we conduct thorough experiments to explore the potential relationship between training efficiency and effectiveness as detailed in Section 3. Subsequently, we extend our findings to larger training configurations of Diffusion Transformers to ascertain the efficacy of our methodologies, as discussed in Section 4.

## A.1 Details of Training

The specific details of the training processes are delineated in Table 4 and Table 5. A notable distinction arises in Section 3, where, to expedite training, we pre-computed and stored image features, abstaining from the use of data augmentation during the training phase. Training was conducted at a resolution of 128, with each experiment running for 100,000 iterations. Conversely, in Section 4, our primary emphasis is on the outcomes of training. Consequently, we opted not to pre-compute image features and instead implemented various image enhancement techniques. Furthermore, the training duration was extended to ensure comprehensive model evaluation.

## A.2 Details of Sampling

In this section, we detail the sampling procedures as discussed in Chapters 3 and 4. The specific outcomes are presented in Table 6 and Table 7. A key distinction between the two sections is as follows: In Section 3, we assessed the FID of 10,000 generated images without employing the *cfg* operation. Conversely, in Section 4, we evaluated the FID of 50,000 generated images and adjusted the *cfg* to 1.5, aligning with the methodologies of prior studies [37, 32].

## A.3 Details of Noise Scheduling

The focus of this paper is on the accelerated training of generative models, with a particular emphasis on Diffusion Transformers [37]. The design of the generative process presented herein draws inspiration from the pioneering work on DiT [37] and the subsequent advancements in SiT [32]. The former primarily adopts the DDPM generation pipeline [22], while the latter enhances DiT training by integrating score-based diffusion [43] and flow matching.

In Section 3, we employ four distinct noise schedules, encompassing both diffusion and flow matching. Typically, these can be represented as $x_t = \alpha_t x_\star + \sigma_t \epsilon$. Specifically, the initial two schedules (Figure 2a&b) employ the widely utilized DDPM linear schedule [22] and the DDPM cosine schedule [35], respectively. Owing to the complexity of these concepts, a detailed discussion is beyond the scope of this paper; readers are encouraged to consult the original publications for more comprehensive information. Regarding the last two (Figure 2c&d), we utilize the two most common flows, as delineated in Equations 9 and 10.

$$\text{Figure 2c} : \alpha_t = t, \sigma_t = 1 - t \tag{9}$$

$$\text{Figure 2d} : \alpha_t = cos(\frac{1}{2}\pi t), \sigma_t = sin(\frac{1}{2}\pi t) \tag{10}$$

# B More Visualization Results

In this section, we present more images generated by FasterDiT, which underwent training on the ImageNet dataset at a resolution of 256 for 1,000,000 iterations. The visualization results, depicted in Figure 9&10, feature samples from seven distinct ImageNet categories, emphasizing selected outputs. During the testing phase, the $cfg$ value was consistently set at 4, in accordance with parameters established in previous research. Remarkably, despite a substantially shorter training period compared to DiT, FasterDiT has successfully generated visually impressive results.

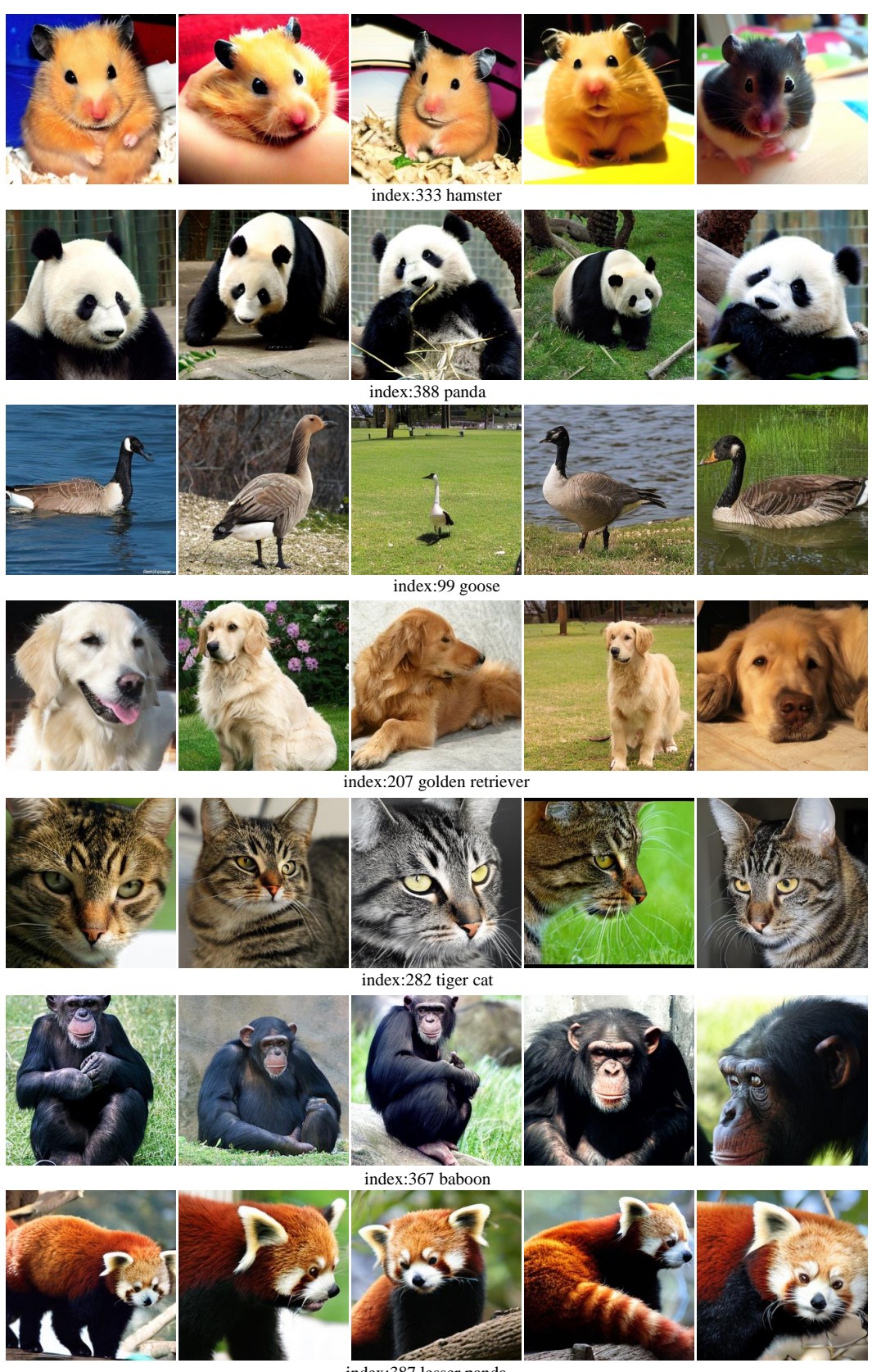

index:333 hamster

index:388 panda

index:99 goose

index:207 golden retriever

index:282 tiger cat

index:367 baboon

index:387 lesser panda

Figure 9: **Generation Results-1.** We visualize generation results of FasterDiT, which is trained on ImageNet at 256 resolution for 1000k iterations.

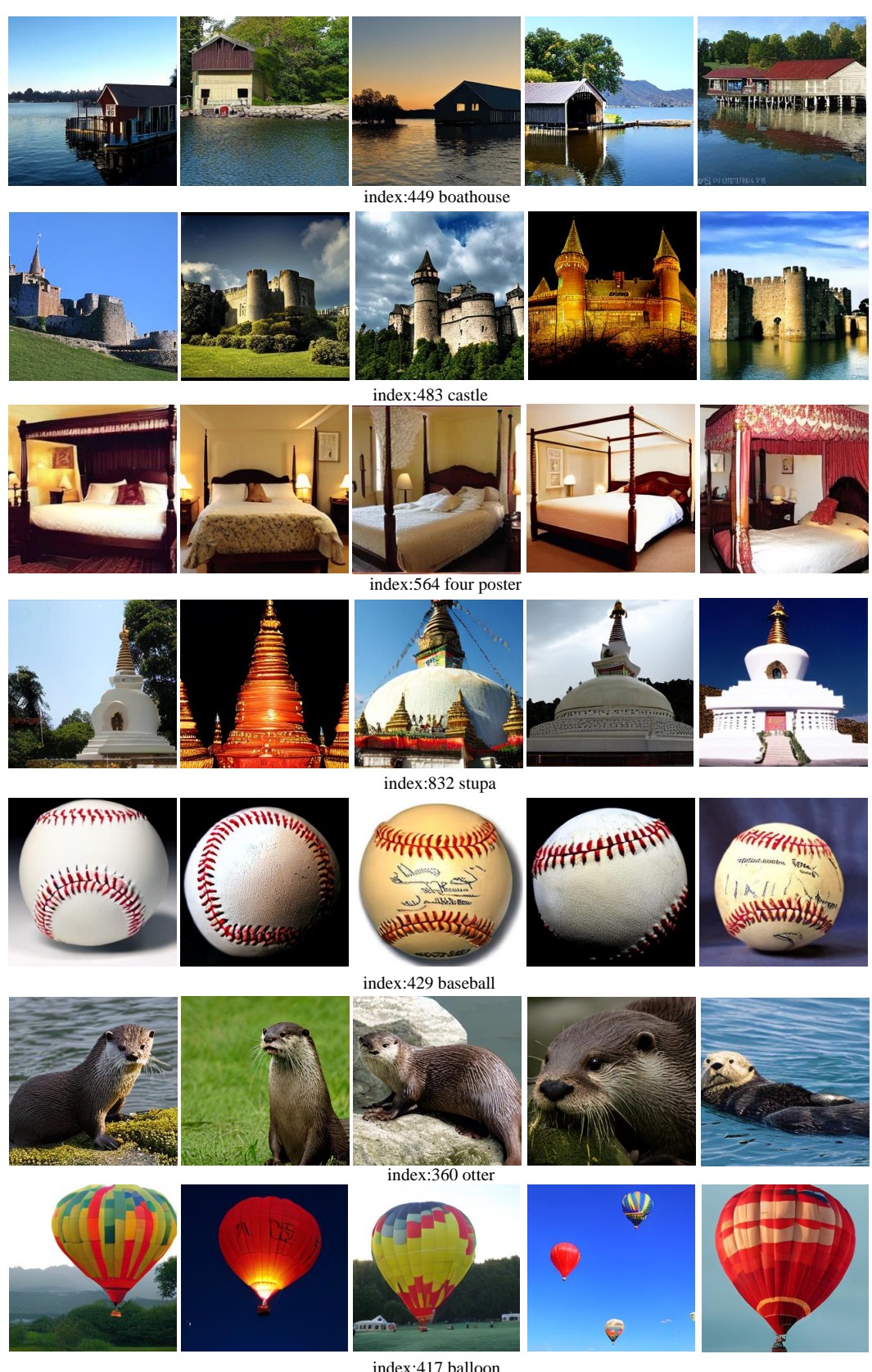

index:449 boathouse

index:483 castle

index:564 four poster

index:832 stupa

index:429 baseball

index:360 otter

index:417 balloon

Figure 10: **Generation Results-2.** We visualize generation results of FasterDiT, which is trained on ImageNet at 256 resolution for 1000k iterations.

Table 4: Training Details of Section 3

| Parameters | Value |
| --- | --- |
| Optimizer | AdamW |
| $\beta_1, \beta_2$ | 0.9, 0.999 |
| Learning Rate | 1e-4 |
| Weight Decay | 0 |
| Global Batchsize | 256 |
| Training Iterations | 100,000 |
| Dataset | ImageNet [13] |
| Resolution | 128 |
| Number Workers | 4 |
| Loss Function | $L_{mse}$ |
| Precompute VAE Features | *yes* |
| Timestep Sampling | *none/ lognorm(0, 1)/ lognorm(0, 0.5)* |
| Data Augmentation | *none* |

Table 5: Training Details of Section 4

| Parameters | Value |
| --- | --- |
| Optimizer | AdamW |
| $\beta_1, \beta_2$ | 0.9, 0.999 |
| Learning Rate | 1e-4 |
| Weight Decay | 0 |
| Global Batchsize | 256 |
| Training Iterations | 1,000,000 |
| Dataset | ImageNet [13] |
| Resolution | 256 |
| Number Workers | 4 |
| Loss Function | $L_{mse}, L_d$ (Eq 8) |
| Precompute VAE Features | *no* |
| Timestep Sampling | *lognorm(0, 1)* |
| Data Augmentation | *Random Horizontal Flip* |

Table 6: Sampling Details of Section 3

| Parameters | Value |
| --- | --- |
| Resolution | 128 |
| Batchsize per GPU | 32 |
| Number of Classes | 1000 |
| *cfg* | 1.0 |
| Number of Samples | 10,000 |
| Number of Sampling Stpes | 250/adaptive |
| Global Seed | 0 |
| Using tf32 | *yes* |

Table 7: Sampling Details of Section 4

| Parameters | Value |
| --- | --- |
| Resolution | 256 |
| Batchsize per GPU | 128 |
| Number of Classes | 1000 |
| *cfg* | 1.5 |
| Number of Samples | 50,000 |
| Number of Sampling Stpes | adaptive |
| Global Seed | 0 |
| Using tf32 | *yes* |

