# OpenReview forum: "FasterDiT: Towards Faster Diffusion Transformers Training without Architecture Modification"
_NeurIPS.cc/2024/Conference — NeurIPS 2024 poster_

### Official Review · Reviewer_FhtE · 2024-07-11

**Soundness:** 3
**Presentation:** 2
**Contribution:** 3
**Rating:** 6
**Confidence:** 5

**Summary:**

The paper presents FasterDiT, a method intended to speed up the training of Diffusion Transformers (DiT) without making changes to the architecture. By utilizing insights from the Probability Density Function (PDF) of Signal-to-Noise Ratio (SNR), FasterDiT improves training strategies and the effectiveness of supervision. The paper also includes a thorough set of experiments and a new supervision approach called FasterDiT.

**Strengths:**

The paper introduces an innovative approach to training strategies by considering the Probability Density Function (PDF) of Signal-to-Noise Ratio (SNR), aiming to improve training efficiency. It includes a thorough empirical analysis with significative experimental results, providing robust evidence to support the findings. FasterDiT achieves significant acceleration in training Diffusion Transformers with a notable increase in training speed, making it a relevant and practical solution for enhancing the efficiency of training large-scale generative models.

**Weaknesses:**

The paper would benefit from additional experiments to demonstrate the generalizability of the proposed approach. A more comprehensive comparison with existing methods for accelerating training of Diffusion Transformers would provide a broader context for evaluating the effectiveness of FasterDiT. Additionally, the scalability of FasterDiT to larger datasets or more complex tasks is not sufficiently discussed, potentially limiting its applicability. The absence of theoretical results and proofs in the paper limits the depth of understanding of the proposed method.

**Questions:**

No questions.

**Limitations:**

A more extensive discussion regarding the drawbacks of the proposed approach is needed.

---

> ### Author Rebuttal · Authors · 2024-08-06
>
> $\color{red}{Question1:}$ The paper would benefit from additional experiments to demonstrate the generalizability of the proposed approach. Additionally, the scalability of FasterDiT to larger datasets or more complex tasks is not sufficiently discussed, potentially limiting its applicability.
>
> $\color{blue}{Response1:}$ Thanks for your suggestion. As requested, we have conducted three types of additional experiments to further explore the generalization of FasterDiT. Please refer to **Response to All Reviewers** for details.
>
> $\color{red}{Question2:}$ A more comprehensive comparison with existing methods for accelerating training of Diffusion Transformers would provide a broader context for evaluating the effectiveness of FasterDiT.
>
> $\color{blue}{Response2:}$ Previous well-known acceleration methods, such as MaskDiT [1], MDT [2], and MDTv2 [3], have achieved significant performance improvements by **modifying architectures** and redesigning the training pipelines of DiT. In contrast, our work approaches the problem from a different perspective. Our aim is to improve DiT training with minimal modifications and **without any changes to the architecture**. This approach is complementary to previous methods. In the future, we plan to explore how combining FasterDiT with more structured improvements can achieve even more efficient generation performance.
>
> $\color{red}{Question3:}$ The absence of theoretical results and proofs in the paper limits the depth of understanding of the proposed method.
>
> $\color{blue}{Response3:}$ Thank you for your suggestions. Our work is to analyze the relationship between SNR PDFs and generation performance through a large number of experiments. We try to derive empirical results and insights to accelerate diffusion training. We will further explore this issue from a theoretical perspective in the future.
>
> $\color{red}{Limitations:}$ A more extensive discussion regarding the drawbacks of the proposed approach is needed.
>
> $\color{blue}{Response:}$ The main limitation of this paper lies in the lack of exploration of large-scale experiments, such as 2K high-resolution images, text-to-image generation, and video generation. Among these, we particularly focus on the text-to-image generation aspect. Specifically, in the class-conditional generation described in this paper, the DiT block only needs to process image tokens. However, in some text-to-image architectures, such as SD3 [4], self-attention needs to handle sequences combining text and visual tokens. The features from different sources (such as T5 and VAE) may lead to potential instability. We plan to further investigate this issue in the future. Besides, we will update our section on 'Limitations' in the next version of our paper. Thanks for your suggestion.
>
> *Refs:*
>
> [1] Zheng H, Nie W, Vahdat A, et al. Fast training of diffusion models with masked transformers[J]. arXiv preprint arXiv:2306.09305, 2023.
>
> [2] Gao S, Zhou P, Cheng M M, et al. Masked diffusion transformer is a strong image synthesizer[C]//Proceedings of the IEEE/CVF International Conference on Computer Vision. 2023: 23164-23173.
>
> [3] Gao S, Zhou P, Cheng M M, et al. MDTv2: Masked Diffusion Transformer is a Strong Image Synthesizer[J]. arXiv preprint arXiv:2303.14389, 2023.
>
> [4] Esser P, Kulal S, Blattmann A, et al. Scaling rectified flow transformers for high-resolution image synthesis[C]//Forty-first International Conference on Machine Learning. 2024.

---

### Official Review · Reviewer_dNCD · 2024-07-13

**Soundness:** 3
**Presentation:** 3
**Contribution:** 3
**Rating:** 6
**Confidence:** 3

**Summary:**

This work aims at solving the slow training convergence of Diffusion Transformer, from the perspective of Signal-to-Noise Ratio (SNR).
Different from other works, the authors formulate the probability density function (PDF) of SNR during training, and then leverage such SNR PDF to analyze the association between training performance and robustness across some common pipelines of DiT.
The finding of the trade-off between method robustness and performance makes authors to propose Fast-DiT which significantly accelerate DiT's training convergence.

**Strengths:**

1. The motivation is very clear. DiT suffers from slow convergence and this work successfully solved this issue without any changes in model architecture.

2. The formulation of SNR PDF is novel. With this analysis tool, authors can check the robustness and performance among various DiT training pipeline.

3. The emipirical experiments are sufficient and the final results compared with SOTA are convincing for me.

**Weaknesses:**

1. Current Table 1 compares the CFG  (classifier-free guidance) results.
Please compare the Class-conditional results of Faster-DiT with DiT and SiT, under the same setting in Table 1 (or even more iterations of Faster DiT).
Usually we need both CFG and Class-conditional results to check the training convergence. Currently Figure 1 right with small iterations is not convincing.

2.  I am still unclear how you change the std of data without changing dataset. Please specify this point.

**Questions:**

Listed in weakness.

**Limitations:**

The authors has discussed the limations of the best performance of FastDiT without sufficient GPUs.

---

> ### Author Rebuttal · Authors · 2024-08-06
>
> $\color{red}{Question1:}$ Current Table 1 compares the CFG (classifier-free guidance) results. Please compare the Class-conditional results of Faster-DiT with DiT and SiT, under the same setting in Table 1 (or even more iterations of Faster DiT). Usually we need both CFG and Class-conditional results to check the training convergence. Currently Figure 1 right with small iterations is not convincing.
>
> $\color{blue}{Response1:}$ Thank you for your suggestion.
> 1. As requested, we have added the following content: (1) We continue training FasterDiT up to 2000k iterations. (2) We additionally report the generation results without CFG (classifier-free guidance). The results indicate that FasterDiT still demonstrates a significant advantage. Please refer to the **Response to All Reviewers** for details.
>
> 2. All experiments in Figure 1 are conducted on ImageNet and trained for 100k iterations. This setting stems from our observation of the training process that the trend of the training FID at 100k is similar to the trend of the training FID over a longer period of time. For example, the table below shows one set of our experiments on ImageNet. It shows that the trend of FID-50k at 100k is similar to the trend of FID-50k at 200k.
>
>     | Type | Std |FID-50 (100k) | FID-50k (200k) |
>     |:------:|:------:|:------:|:------:|
>     |DDPM cosine schedule| 0.5 | 43.22 | 22.08 |
>     |  | 0.6 | 61.15 | 25.70 |
>     |  | 0.7 | 64.18 | 25.96 |
>     |  | 0.8 | 72.39 | 28.94 |
>     |  | 0.9 | 78.45 | 30.45 |
>     |  | 1.0 | 98.91 | 42.41 |
>     |  | 1.1 | 81.26 | 38.10 |
>     |  | 1.2 | 111.83 | 42.44 |
>
> $\color{red}{Question2:}$ I am still unclear how you change the std of data without changing dataset. Please specify this point.
>
> $\color{blue}{Response2:}$ The $std$ here refers to the standard deviation of the input. We can achieve the scaling of the standard deviation simply by multiplying a scale factor.

---

> > ### Comment · Reviewer_dNCD · 2024-08-12
> >
> > Thanks for your experiments and answers! These answers have solved my problems.
> > I will keep my ratings.

---

### Official Review · Reviewer_ttWn · 2024-07-13

**Soundness:** 3
**Presentation:** 1
**Contribution:** 3
**Rating:** 6
**Confidence:** 3

**Summary:**

This paper propose FasterDiT, a diffusion model training method that considers the data distribution in the definition of signal-to-noise ratio. It formulates the SNR in a new framework, estimates the PDF of the SNR, and then employs it to improve the training efficiency of DiT. Experimental results show that FasterDiT achieves relative FID but with 1/7 training time.

**Strengths:**

1. The idea of re-formulating the SNR in the diffusion model is a very interesting and novel strategy for me. I believe this paper has a good technical contribution.
2. The experimental results of DiT show both good acceleration and generation performance.

**Weaknesses:**

1. This paper has a bad quality in writing. There are some typos and the explanation to the SNR is not clear to me. I'm confused for many questions: (1) What does the std means in Line 109. Does it mean the std of the value of pixels in the images?  (2) Why can we assume std^2 approximate K(I)/std^2 as a constant C(I)?  (3) Authors mention "robustness" for many times in the paper for "training robustness", "data robustness". What does it mean in detail? (4) In Figure 6, the caption inside the figure is "multi-step balance", while the caption after the figure is multiple-step balance. Please use the same description. What does multi-step balance mean? Does that mean the new training strategy with SNR?  The "single-step supervision" is also not a good choice here. It should be something like "directionality of velocity" or "single-step supervision (ours)" since there has already been single-step supervision in the traditional training method.
2. Does the proposed method generalize well to the diffusion models besides DiT, such as latent diffusion models? Please discuss on this. If so, experimental results should be provided for this since only experiments on DiT & ImageNet is not very convincing.

**Questions:**

1. Does x_* in the paper indicate the x_0 (real images)? If so, I advise to replace it with x_0 to align with previous works.
2. Section 4: Improving DiT Training. The "." should be removed.
3. The space between line 140 and 141 are excessively reduced.
In summary, I think this paper may have good technical contribution. But it's really difficult for me to understand the details of this paper.

---

> ### Author Rebuttal · Authors · 2024-08-06
>
> $\color{red}{Question1:}$ What does the $std$ means in Line 109. Does it mean the $std$ of the value of pixels in the images?
>
> $\color{blue}{Response1:}$ The $std$ here refers to the standard deviation of the input. For traditional diffusion, it refers to the standard deviation of pixel values. For latent diffusion, it refers to the standard deviation of the latents after VAE encoding. In this paper, our experiments are primarily based on latent diffusion.
>
> $\color{red}{Question2:}$ Why can we assume $std^2$ approximate $K(I)/std^2$ as a constant $C(I)$?
>
> $\color{blue}{Response2:}$ The approximation maintains our definition as an extension of the previous one with minimal changes. Specifically, in previous definitions, the standard deviation ($std$) is the most crucial factor in quantifying the Signal-to-Noise Ratio ($SNR$). For ease of analysis, previous works [1, 4] treat both the input and noise as normal distributions, resulting in:
>
> $SNR_{pre} = \frac{(\alpha_t \times std_{input})^2}{(\sigma_t \times std_{noise})^2}=\frac{\alpha_t^2}{\sigma_t^2}, std_{input}=std_{noise}=1$
>
> In FasterDiT, our definition extends this from the following two aspects: Firstly, we take the standard deviation of input back into consideration instead of assuming it to be one. Secondly, some work [5] has shown that $SNR$ relates to more properties of inputs besides $std_{input}$ such as resolution. Here, we denote these additional properties as $K(I)/std_{input}^2$. Thus, we obtain:
>
> $SNR = \frac{(\alpha_t \times std_{input})^2}{\sigma_t^2}\times\frac{K(I)}{std_{input}^2}$
>
> In our work, we pay more attention to the former aspect and relax the definition. We assume the $std_{input}^2$ is a decoupled parameter in $K(I)$ for ease of analysis similar to the previous definition. It helps us simplify the modeling of the SNR PDF with minimal impact on the qualitative results. We plan to conduct a more in-depth exploration of this in our future work.
>
> $\color{red}{Question3:}$ Authors mention "robustness" for many times in the paper for "training robustness", "data robustness". What does it mean in detail?
>
> $\color{blue}{Response3:}$ Here, "robustness" refers to the stability of training performance despite variations in the standard deviation of the input data. We will standardize this terminology in future versions.
>
> $\color{red}{Question4:}$ In Figure 6, the caption inside the figure is "multi-step balance", while the caption after the figure is multiple-step balance. Please use the same description. What does multi-step balance mean? Does that mean the new training strategy with SNR? The "single-step supervision" is also not a good choice here. It should be something like "directionality of velocity" or "single-step supervision (ours)" since there has already been single-step supervision in the traditional training method.
>
> $\color{blue}{Response4:}$ The "multi-step balance" refers to modifying the SNR PDF by adjusting the proportion of different timesteps during training. We will standardize the terminology and improve the naming of the direction loss to enhance clarity. Thank you for your suggestions.
>
> $\color{red}{Question5:}$ Does the proposed method generalize well to the diffusion models besides DiT, such as latent diffusion models? Please discuss on this. If so, experimental results should be provided for this since only experiments on DiT & ImageNet is not very convincing.
>
> $\color{blue}{Response5:}$ Thanks for the suggestion. Yes, it does. The method proposed in FasterDiT is decoupled from the model architecture. It could be interesting to explore its performance alongside DiT. As requested, we employed the U-Net used in LDM [2] and another well-known architecture, U-ViT [3], to investigate the influence of our method. The results demonstrate that FasterDiT continues to enhance the convergence rate of these models. Please refer to **Response to All Reviewers** for more details.
>
> $\color{red}{Question6:}$ (1) Does $x_*$ in the paper indicate the $x_0$ (real images)? If so, I advise to replace it with $x_0$ to align with previous works. (2) Section 4: Improving DiT Training. The "." should be removed. (3) The space between line 140 and 141 are excessively reduced. In summary, I think this paper may have good technical contribution. But it's really difficult for me to understand the details of this paper.
>
> $\color{blue}{Response6:}$ Thank you very much for your careful review of our submission and your detailed suggestions. In this paper, we explore DiT training with both diffusion and flow matching. We didn't choose $x_0$ to denote the input because of the different timestep definitions. Different from DDPM, in Rectified Flow, $x (t=0)$ refers to the pure noise. We will consider clearer expressions in future versions of our paper. Besides, we will address all the other issues you raised and thoroughly review the entire manuscript to prevent similar issues from occurring in the future.
>
> *Refs:*
>
> [1] Salimans T, Ho J. Progressive Distillation for Fast Sampling of Diffusion Models[C]//International Conference on Learning Representations.
>
> [2] Rombach R, Blattmann A, Lorenz D, et al. High-resolution image synthesis with latent diffusion models[C]//Proceedings of the IEEE/CVF conference on computer vision and pattern recognition. 2022: 10684-10695.
>
> [3] Bao F, Nie S, Xue K, et al. All are worth words: A vit backbone for diffusion models[C]//Proceedings of the IEEE/CVF conference on computer vision and pattern recognition. 2023: 22669-22679.
>
> [4] Hang T, Gu S, Li C, et al. Efficient diffusion training via min-snr weighting strategy[C]//Proceedings of the IEEE/CVF International Conference on Computer Vision. 2023: 7441-7451.
>
> [5] Hoogeboom E, Heek J, Salimans T. simple diffusion: End-to-end diffusion for high resolution images[C]//International Conference on Machine Learning. PMLR, 2023: 13213-13232.

---

> ### Comment · Reviewer_ttWn · 2024-08-13
> **Response to author rebuttal**
>
> After reading the response from the authors, I decide to increase my rate from 5 to 6. I believe this paper has real novelty in the training of diffusion models and it still can be greatly improved by better writing. Please keep polishing it.

---

### Official Review · Reviewer_qPr3 · 2024-07-13

**Soundness:** 2
**Presentation:** 2
**Contribution:** 3
**Rating:** 6
**Confidence:** 3

**Summary:**

The paper focuses on accelerating the training process of Diffusion Transformers (DiT) without modifying their architecture. The authors identify two primary issues: inconsistent performance of certain training strategies across different datasets, and limited effectiveness of supervision at specific timesteps.

Key contributions include:
1. Extended Definition of Signal-to-Noise Ratio (SNR)
2. Extensive experiments and empirical findings for SNR PDF
3. A new supervision method

**Strengths:**

- FasterDiT achieves competitive results (2.30 FID on ImageNet 256 resolution at 1000k iterations) while being seven times faster in training compared to traditional DiT (2.27 FID).
- The paper presents a large number of experiments to empirically validate the proposed method. The paper presents various discussions and insights of SNR PDF.
- By generalizing the definition of SNR and analyzing various training strategies through SNR PDFs, the paper shows promising outcomes of faster training.

**Weaknesses:**

- The experiments were conducted on 256-resolution ImageNet only. It would be interesting to validate the proposed method on larger resolutions (such as 512, 1024, etc.). Acceleration is more critical to those scenarios.
- The figures could be improved. The rendered text is not easy to read.

**Questions:**

see weakness

---

> ### Author Rebuttal · Authors · 2024-08-06
>
> $\color{red}{Question1:}$ The experiments were conducted on 256-resolution ImageNet only. It would be interesting to validate the proposed method on larger resolutions (such as 512, 1024, etc.). Acceleration is more critical to those scenarios.
>
> $\color{blue}{Response1:}$ Thanks for your suggestion. As requested, we have conducted experiments with higher resolution (ImageNet 512x512). It shows that FasterDiT could still achieve faster convergence speed compared to the original DiT. Please refer to our **Response to All Reviewers** for more details. Due to the limited time available for rebuttal, we will explore higher resolution and larger training scales in the future.
>
> $\color{red}{Question2:}$ The figures could be improved. The rendered text is not easy to read.
>
> $\color{blue}{Response2:}$ Thank you very much for your kind suggestions. We will improve our figures in the future version of our manuscript.

---

> > ### Comment · Reviewer_qPr3 · 2024-08-10
> > **comment**
> >
> > Thank you for the experiments. I hope the authors will add the high-res results to the final paper.

---

### Official Review · Reviewer_fAgp · 2024-07-23

**Soundness:** 3
**Presentation:** 3
**Contribution:** 3
**Rating:** 6
**Confidence:** 3

**Summary:**

This paper presents observations on training strategies for Diffusion Transformers (DiT) using Signal-to-Noise Ratio (SNR) Probability Density Function (PDF) analysis. Through extensive experiments, the authors derive insights into training performance and robustness. Based on these findings, they propose a method to accelerate the training process of DiT without modifying the model architecture.

**Strengths:**

1. The problem addressed is practical, focusing on the significant computational demands of training Diffusion Transformer models.
2. The authors propose a simple yet effective technique to improve the training efficiency of Diffusion Transformer models without modifying the architecture
3. The paper is well-structured with clear motivation, methodology, and results.

**Weaknesses:**

1. The paper lacks experiments with higher resolutions such as 512x512 or 1024x1024, which could provide insights into the method's scalability for more complex image generation tasks.
2. The hyperparameter 'std' requires tuning to find the appropriate setting, which may introduce additional complexity in implementing the method across different datasets or tasks.

**Questions:**

Why does adding cosine similarity loss for velocity direction supervision accelerate the convergence rate of the model?

**Limitations:**

The scalability of this method is not thoroughly evaluated, including its effectiveness for higher resolutions, other datasets, or different types of generative tasks (e.g., text-to-image, video generation).

---

> ### Author Rebuttal · Authors · 2024-08-06
>
> $\color{red}{Question1:}$ The paper lacks experiments with higher resolutions such as 512x512 or 1024x1024, which could provide insights into the method's scalability for more complex image generation tasks.
>
> $\color{blue}{Response1:}$ Thanks for the suggestion. As requested, we have conducted experiments on a higher resolution (ImageNet 512x512) using two different model scales (DiT-B and DiT-L). The results show that FasterDiT still achieves faster convergence at higher resolutions. For more details, please refer to our **Response to All Reviewers**.
>
> $\color{red}{Question2:}$ The hyperparameter 'std' requires tuning to find the appropriate setting, which may introduce additional complexity in implementing the method across different datasets or tasks.
>
> $\color{blue}{Response2:}$ Thank you for the question. We will explain it from two perspectives.
>
> Firstly, we believe that an appropriate standard deviation (std) of the diffusion input can influence training performance in certain cases. This observation is similar to the concept of a 'magic number' or 'scale factor' in previous well-known studies [1, 2]. Using the CIFAR10 toy setting as an example, when we apply the most common image normalization, the standard deviation (std) of the data is approximately 0.4, and the training FID result is 51.40. However, when we adjust the std to 0.8, the result of the same training method improves significantly to 36.31. This demonstrates the importance of std adjustment.
>
> | Model | Std |Training Samples | FID-10k |
> |:--------|:--------:|:--------:|:--------|
> | DiT-S/2 | 0.4 | 50k x 128 | 51.40 |
> | DiT-S/2 | 0.8 | 50k x 128 | 36.31 (-15.09) |
>
> Secondly, the complexity brought by tuning the 'std' will decrease as research on diffusion deepens. For instance, [2] suggested that as resolution increases, training tends to favor a smaller std. In our work, the SNR PDF provides a new analytical tool and offers insights into this issue. We hope it can facilitate a deeper understanding of the diffusion process.
>
> $\color{red}{Question3:}$ Why does adding cosine similarity loss for velocity direction supervision accelerate the convergence rate of the model?
>
> $\color{blue}{Response3:}$ The direction loss of velocity serves as an auxiliary form of supervision that enables a more fully supervised training process. To provide further insights, we have conducted an ablation study. We train two DiT-B models: one with FasterDiT and the other with FasterDiT without direction loss. We then visualize the training loss at different timestep intervals, which can be seen in our **submitted PDF**. The results show that the model with directional loss demonstrates a significantly better training effect, particularly during the diffusion period of relative low-noise ($0.4 < t < 0.9$). The end-to-end performance comparison is shown in the table below.
> | Method | Models | Training Samples | Resolution | FID-10k (100k) | FID-10k (200k) |
> |:--------|:--------:|:--------:|:--------:|:--------:|:--------:|
> | FasterDiT (w/o direction loss)    | DiT-B/2 | 100k x 128 | 512x512 | 83.29 | 63.11 |
> | FasterDiT     | DiT-B/2 | 100k x 128 | 512x512 | **77.85** | **58.33** |
>
> $\color{red}{Limitations}:$ The scalability of this method is not thoroughly evaluated, including its effectiveness for higher resolutions, other datasets, or different types of generative tasks (e.g., text-to-image, video generation).
>
> $\color{blue}{Response:}$ Thanks for your suggestion. We have provided experiments with higher resolutions and different models of FasterDiT. We will explore more complex tasks you mentioned such as T2I generaiton and video generation in our following work.
>
> *Refs:*
>
> [1] Rombach R, Blattmann A, Lorenz D, et al. High-resolution image synthesis with latent diffusion models[C]//Proceedings of the IEEE/CVF conference on computer vision and pattern recognition. 2022: 10684-10695.
>
> [2] Chen T. On the importance of noise scheduling for diffusion models[J]. arXiv preprint arXiv:2301.10972, 2023.

---

> > ### Comment · Reviewer_fAgp · 2024-08-12
> >
> > Thank you for your response. These additional experiments address most of my concerns. Based on this, I am willing to upgrade my score.

---

### Author Rebuttal · Authors · 2024-08-06

## Response to All Reviewers

We appreciate all of your valuable feedback. Your valuable suggestions have significantly improved our manuscript.

All reviewers have suggested that we conduct additional experiments to demonstrate the effectiveness and generalization ability of our method. As requested, we have added the following **three types of additional experiments**:
1. Training FasterDiT with longer iterations and performance without CFG (R_dNCD).
2. Training FasterDiT with higher resolution (R_fAgp, R_qPr3)
3. Applying the FasterDiT strategy to different diffusion models (R_ttWn, R_FhtE).

Due to the limited time available for the rebuttal, we have conducted experiments to the maximum extent that our resources allow. The larger scale experiments mentioned by the reviewers, such as 1024 resolution, text-to-image generation ,and video generation, cannot be completed within this short period. We plan to explore these in future work. Thank you for all the suggestions.

### 1. Longer Training of FasterDiT:

We further improve the performance of FasterDiT with longer training iterations and report more results of FID-50k on ImageNet 256 resolution.

It shows that even without any structural modifications, FasterDiT could achieve comparable results to the original DiT at 1000k iterations. Furthermore, when the training period is extended to 2000k iterations, FasterDiT's performance further improves, achieving an FID-50k score of 2.03, demonstrating the effectiveness of our approach.

**Performance without Classifier-Free Guidance (CFG)**

| Method     | Models     | Training Samples     | FID-50k   |
|:--------|:--------:|:--------:|:--------:|
| DiT     | DiT-XL/2 | 7000k x 256 | 9.60 |
| SiT    | DiT-XL/2 | 7000k x 256 | 8.60 |
| FasterDiT    | DiT-XL/2 | 1000k x 256 | 8.72 |
|     |  | 1500k x 256 | 8.22 |
|     |  | **2000k x 256** | **7.91** |

**Performance with Classifier-Free Guidance (CFG)**

| Method     | Models     | Training Samples     | FID-50k   |
|:--------|:--------:|:--------:|:--------:|
| DiT (*cfg=1.5*)     | DiT-XL/2 | 7000k x 256 | 2.27 |
| SiT (*cfg=1.5*)     | DiT-XL/2 | 7000k x 256 | 2.06 |
| FasterDiT (*cfg=1.5*)    | DiT-XL/2 | 1000k x 256 | 2.30 |
|     |  | 1500k x 256 | 2.12 |
|     |  | **2000k x 256** | **2.03** |

### 2. Higher Resolution:

As requested, we explore FasterDiT on higher resolution generation experiments to demonstrate its generalization ability. Specifically, we apply our method to DiT-B/2 and DiT-L/2 for ImageNet 512x512 generation training. The results are shown in the table below.

FasterDiT achieves faster convergence across all experiments. When trained with 200k iterations, we improve the FID-10k performance of DiT-B/2 by 18.78 and DiT-L/2 by 17.93. This demonstrates the effectiveness of our method in high-resolution tasks.

| Method | Models | Training Samples | Resolution | FID-10k   |
|:--------|:--------:|:--------:|:--------:|:--------|
| DiT     | DiT-B/2 | 100k x 128 | 512x512 | 93.36 |
|      |  | 200k x 128 | 512x512 | 77.11 |
| FasterDiT     | DiT-B/2 | 100k x 128 | 512x512 | 77.85 (-11.51)  |
|      |  | 200k x 128 | 512x512 | 58.33 **(-18.78)** |
| DiT     | DiT-L/2 | 100k x 64 | 512x512 | 87.24 |
|      |  | 200k x 64 | 512x512 | 67.29 |
| FasterDiT     | DiT-L/2 | 100k x 64 | 512x512 | 71.58 (-15.66)  |
|      |  | 200k x 64 | 512x512 | 49.36 **(-17.93)** |


### 3. Different Architectures:

As requested, we explore our training method with other diffusion models besides DiT, such as Latent Diffusion Models (UNet architecture) [1] and U-ViT [2]. The results are shown in the table below.


It shows that with our training methods, the performance of U-ViT and UNet both improve. It demonstrates that our method might have opportunities to be generalized to more diffusion architectures.


| Model | Training Samples | FID-10k |
|:--------|:--------:|:--------:|
| U-ViT-L |  200k x 128 | 50.22 |
| U-ViT-L + Ours |  200k x 128 | **37.12** |
| UNet | 200k x 128 | 66.73 |
| UNet + Ours | 200k x 128 | **60.07**


*Refs:*

[1] Rombach R, Blattmann A, Lorenz D, et al. High-resolution image synthesis with latent diffusion models[C]//Proceedings of the IEEE/CVF conference on computer vision and pattern recognition. 2022: 10684-10695.

[2] Bao F, Nie S, Xue K, et al. All are worth words: A vit backbone for diffusion models[C]//Proceedings of the IEEE/CVF conference on computer vision and pattern recognition. 2023: 22669-22679.

---

### Decision · Program_Chairs · 2024-09-25

**Decision:**

Accept (poster)

**Comment:**

The paper introduces an innovative approach to accelerate the training of Diffusion Transformers (DiT). The authors present a thorough empirical analysis, validating their method across various experiments and demonstrating competitive performance on ImageNet, while being significantly faster than the traditional DiT. The reviewers were unanimous in the decision to accept the paper and the AC agrees with the decision. The AC recommends that the authors put extra effort in polishing the paper, as well as adding the additional experiments provided in rebuttal.